# Characteristics and Comparative Analysis of Seven Complete Plastomes of *Trichoglottis* s.l. (Aeridinae, Orchidaceae)

**DOI:** 10.3390/ijms241914544

**Published:** 2023-09-26

**Authors:** Cheng-Yuan Zhou, Meng-Yao Zeng, Xuyong Gao, Zhuang Zhao, Ruyi Li, Yuhan Wu, Zhong-Jian Liu, Diyang Zhang, Ming-He Li

**Affiliations:** 1Key Laboratory of National Forestry and Grassland Administration for Orchid Conservation and Utilization at Landscape Architecture and Arts, Fujian Agriculture and Forestry University, Fuzhou 350002, China; zcy810338055@126.com (C.-Y.Z.); zmy13860927342@163.com (M.-Y.Z.); 18050266127@126.com (X.G.); fafuzzhuang@163.com (Z.Z.); ichhabeihn@163.com (R.L.); 15280838312@163.com (Y.W.); zjliu@fafu.edu.cn (Z.-J.L.); 2Fujian Colleges and Universities Engineering Research Institute of Conservation and Utilization of Natural Bioresources, Fujian Agriculture and Forestry University, Fuzhou 350002, China

**Keywords:** *Trichoglottis*, *Staurochilus*, Orchidaceae, plastid genome, phylogenetic analysis

## Abstract

*Trichoglottis* exhibits a range of rich variations in colors and shapes of flower and is a valuable ornamental orchid genus. The genus *Trichoglottis* has been expanded by the inclusion of *Staurochilus*, but this *Trichoglottis sensu lato* (s.l.) was recovered as a non-monophyletic genus based on molecular sequences from one or a few DNA regions. Here, we present phylogenomic data sets, incorporating complete plastome sequences from seven species (including five species sequenced in this study) of *Trichoglottis* s.l. (including two species formerly treated as *Staurochilus*), to compare plastome structure and to reconstruct the phylogenetic relationships of this genus. The seven plastomes possessed the typical quadripartite structure of angiosperms and ranged from 149,402 bp to 149,841 bp with a GC content of 36.6–36.7%. These plastomes contain 120 genes, which comprise 74 protein-coding genes, 38 tRNA genes, and 8 rRNA genes, all *ndh* genes were pseudogenized or lost. A total of 98 (*T*. *philippinensis*) to 134 (*T*. *ionosma*) SSRs and 33 (*T*. *subviolacea*) to 46 (*T*. *ionosma*) long repeats were detected. The consistent and robust phylogenetic relationships of *Trichoglottis* were established using a total of 25 plastid genomes from the Aeridinae subtribe. The genus *Trichoglottis* s.l. was strongly supported as a monophyletic group, and two species formerly treated as *Staurochilus* were revealed as successively basal lineages. In addition, five mutational hotspots (*trnN^GUU^*-*rpl32*, *trnL^UAA^*, *trnS^GCU^*-*trnG^UCC^*, *rbcL*-*accD*, and *trnT^GGU^*-*psbD*) were identified based on the ranking of PI values. Our research indicates that plastome data is a valuable source for molecular identification and evolutionary studies of *Trichoglottis* and its related genera.

## 1. Introduction

*Trichoglottis* Blume is a genus of Aeridinae within the family Orchidaceae, which was established by Carl Blume in 1825 with the type species *Trichoglottis retusa* Blume [1], comprising approximately 85 species [2]. Members of *Trichoglottis* are mainly distributed in tropical Asia and the northwest Pacific [3]. Many species are narrow endemics, and there is a major center of diversity in Indonesia and the Philippines [3,4]. Species of *Trichoglottis* are mainly characterized by monopodial growth, an axillary inflorescence with one or several brightly colored flowers, and a hairy strap-shaped lip firmly fused to a column [4,5]. *Trichoglottis* exhibits a range of rich variations in colors and shapes of flower and is a valuable ornamental orchid. To date, approximately 75 artificial interspecific hybrids have been produced and registered with the Royal Horticultural Society (http://apps.rhs.org.uk/horticulturaldatabase/orchidregister/orchidregister.asp, accessed on 10 July 2023).

Since the establishment of *Trichoglottis*, the intergeneric phylogenetic relationships of this genus have remained unclear and are generally considered to be one of the most complicated groups in Aeridinae. Molecular phylogenetics, based on two markers (*matK* and ITS), indicated that *Trichoglottis* is sister to *Staurochilus*, and both together are sisters to *Ceratochilus* and *Ventricularia* [6]. *Trichoglottis* has been expanded by the inclusion of *Ceratochilus*, *Staurochilus*, and *Ventricularia* [3]. However, the subject of *Staurochilus* versus *Trichoglottis* is still debated. Based on three combined markers (ITS, *matK*, *trnL-F*), *Staurochilus* was revealed as a sister to *Acampe* [7]. Additionally, based on five combined markers (ITS, *atpI-H*, *matK*, *psbA-trnH*, and *trnL-F*), the results showed that *Trichoglottis* was polyphyletic, *Staurochilus* (including *S*. *luchuensis* and *S*. *ionosma*) was paraphyletic, and *T*. *latisepala* was embedded in *Staurochilus* [8]. Nevertheless, many of the relationships among these clades/genera remain uncertain due to weak support. Briefly, traditional plant classification methods based on morphological features are not always adequate for the delimitation of *Trichoglottis*. Furthermore, the use of DNA barcoding for analysis has limitations in terms of resolution, making it difficult to differentiate between *Trichoglottis* and *Staurochilus*.

One traditional approach for increasing the resolution in molecular phylogenetic data sets is to sample more loci [9,10]. Plastids are usually uniparentally inherited and have their own genomes with dense gene content and a moderate mutation rate [11]. The complete plastid genome (plastome) data have proved to be effective tools for orchid species delimitation and molecular marker development [12,13,14,15]. Here, we report new complete plastomes for five *Trichoglottis* species. These new sequences are compared with the two previously described *Trichoglottis* plastomes [15], which (we refer to *Trichoglottis* s.l.) include two species (*T. ionosma* and *T. philippinensis*) formerly placed in *Staurochilus*. Simple sequence repeats (SSRs) were used to investigate the genetic diversity of these plastomes. Codon usage analysis was conducted to find the codon bias. The sequence divergence was calculated for barcoding investigation. Further, a total of 25 previously sequenced plastomes representing all previously recognized major lineages of Aeridinae were integrated for phylogenetic analysis with the aim of understanding the generic delimitation of *Trichoglottis* s.l. The present results provide a useful genetic resource for molecular identification and evolutionary studies of *Trichoglottis* and its related genera.

## 2. Results

### 2.1. Characteristics of the Plastome

The plastomes of *Trichoglottis* displayed a typical quadripartite structure (Figure 1), consisting of a pair of IRs (25,775–25,812 bp), a LSC region (85,681–86,210 bp), and a SSC region (12,004–12,171 bp) (Table 1). The size of the plastomes varied from 149,402 bp (*T*. *latisepala*) to 149,841 bp (*T*. *ionosma*), which fell within the typical range observed for common angiosperm plastome sizes. The GC content of whole plastomes exhibited minimal variation. The *T*. *ionosma* had a content of 36.6%, and the remaining six species exhibited a content of 36.7%.

The plastomes of *Trichoglottis* encoded 120 genes, comprising 74 protein-coding genes, 38 transfer RNA (tRNA) genes, and 8 ribosomal RNA (rRNA) genes (Table 1). All the plastomes had undergone loss or pseudogenization of the *ndh* genes (Figure 1, Table 1). The plastome of *T*. *ionosma* possessed seven pseudogenes (*ndhB*/*C*/*D*/*E/G*/*J*/*K*), the other six species possessed eight pseudogenes (*ndhB*/*C*/*D*/*E*/*G*/*I*/*J*/*K*). The result of Mauve analysis revealed no significant rearrangements among these plastomes (Appendix A).

Visualization of boundary genes in the plastomes revealed a highly conserved distribution pattern among *Trichoglottis* (Appendix A). The *rpl22* gene in all species spanned from LSC to IRb, with a consistent length of 31 bp. At the junction between SSC and IRb (JSB), the *ycf1* gene of *T*. *latisepala* and *T*. *philippinensis* was entirely located within IRb, while, for the other five species, the *ycf1* gene spanned from IRb to SSC, with a range of 11 to 41 bp. In all species, the *ycf1* gene spanned the junction between the SSC and IRa (JSA), and the *trnH* and *psbA* genes were found adjacent to the junction between the IRa and LSC (JLA).

### 2.2. Repeated Analysis

The different types of SSR (mononucleotide, dinucleotide, trinucleotide, tetranucleotide, pentanucleotide, and hexanucleotide) were analyzed in the seven plastomes of *Trichoglottis*. A total of 98 (*T*. *philippinensis*) to 134 (*T*. *ionosma*) SSRs were detected (Figure 2, Appendix A). Mononucleotide repeats were the most frequent type of SSRs (50–90), followed by dinucleotide repeats (14–20). Hexanucleotide occurred with the lowest frequency, and it was exclusively detected in *T*. *subviolacea* (2). Additionally, it was observed that A/T (48–90) dominated over C/G (0–2) in mononucleotide repeats. In dinucleotide repeats, the number of AT/AT (10–16) exceeded that of AG/CT (4). Furthermore, trinucleotide repeats exclusively contained A/T.

Four types of long repeats (palindrome, forward, reverse, and completement) were also detected in *Trichoglottis* plastomes. The number of long repeats ranged from 33 (*T*. *subviolacea*) to 46 (*T*. *ionosma*) (Figure 2, Appendix A). Except for *T*. *subviolacea*, which only possessed three types of long repeats (palindrome, forward, and reverse), the other six species possessed all four types of long repeats. The length of long repeats in all species mostly ranged from 30 bp to 40 bp. For long repeats above 50 bp, only *T*. *ionosma* had one, while the other species possessed two. Complement repeats were only detected below 40 bp in length.

### 2.3. Codon Usage Analyses

The concatenated sequences of 68 protein-coding genes were used to calculate the codon usage frequency of *Trichoglottis* plastomes. These protein-coding genes encoded 19,285–19,351 codons. The heatmap showed that the codon usage bias was highly conserved (Appendix A). The range of relative synonymous codon usage (RSCU) values was observed to be 0.346–1.894, with 30 codons having RSCU values greater than 1. Among these codons, leucine (Leu) had the highest number of amino acids, and cysteine (Cys) had the lowest frequency (Appendix A). The GCU codon had the highest RSCU value (1.881–1.894), while GAC exhibited the lowest RSCU value (0.348–0.353). Among the three types of termination codons (UAA, UAG, UGA), the UAA codon had the highest RSCU values, ranging from 1.412 to 1.456.

### 2.4. Plastome Sequence Divergence and Barcoding Investigation

To assess the variations of *Trichoglottis* plastomes, the mVISTA was employed to investigate the structural characteristics of seven plastomes with the reference *Thrixspermum centipeda*. Appendix A shows that the coding regions were highly conserved in comparison to the non-coding regions. The IR regions were more conserved than the LSC and SSC regions.

In order to explore the highly mutated hotspots of *Trichoglottis* plastomes, the DnaSP6 was employed to calculate nucleotide diversity (Pi). The results showed Pi values ranging from 0 to 0.12857 (*trnN^GUU^*-*rpl32*) (Figure 3A, Appendix A). The IR regions were relatively conserved (Pi = 0.0031) and high levels of nucleotide diversity were detected in the SSC region (Pi = 0.01058), followed by the LSC region (Pi = 0.0052). According to the ranking of Pi values, five hypervariable regions, including *trnN^GUU^*-*rpl32*, *trnL^UAA^*, *trnS^GCU^*-*trnG^UCC^*, *rbcL*-*accD*, and *trnT^GGU^*-*psbD*, were identified. Protein-coding genes were also employed for the analysis of nucleotide diversity and showed high conservation (Figure 3B, Appendix A). Among these genes, *psbK* had the highest Pi, but with only 0.0109.

### 2.5. Phylogenetic Analysis

The phylogenetic trees constructed based on complete plastomes and 68 protein-coding genes revealed similar topological structures, and the main clades were recovered with high-level support (Figure 4, Appendix A). The species of *Trichoglottis* formed a well-supported monophyletic group (BS = 100, PP = 1.00) and was revealed as a sister to *Acampe*. The phylogenetic relationship inferred by complete plastomes showed strong support within *Trichoglottis* (BS ≥ 98, PP = 1.00), but relatively moderate support was shown in trees inferred by 68 protein-coding genes (BS_ML_ ≥ 72, BS_Mp_ ≥ 43, PP ≥ 0.71). The phylogenetic trees showed that *Trichoglottis* could be divided into four diverging lineages. *T*. *ionosma* (formerly treated as *Staurochilus ionosmus*) was supported as the first lineage. *T*. *philippinensis* (formerly treated as *S. philippinensis*) was supported as the second lineage. *T*. *subviolacea* was supported as the third lineage. Four species, *T*. *lanceolaria*, together with *T*. *latisepala*, *T*. *cirrhifera*, and *T*. *orchidea*, were supported as the fourth lineage. Extremely short branch lengths were observed within the fourth lineage.

## 3. Discussion

### 3.1. The Plastome Characteristics and Structural Evolution

In this study, we expanded plastome sampling in *Trichoglottis* and provided a valuable opportunity to further understand plastome evolution in this complex taxon. All plastomes of *Trichoglottis* (Figure 1) displayed the typical quadripartite structure, consisting of one LSC region, one SSC region, and two IR regions, consistent with that of most angiosperm plastomes. The length of *Trichoglottis* plastomes (Table 1) ranged from 149,402 bp (*T*. *latisepala*) to 149,841 bp (*T*. *ionosma*), falling entirely within the reported range of plastomes in Orchidaceae, which ranged from 19,047 bp (*Epipogium roseum*) to 212,688 bp (*Cypripedium tibeticum*) [16,17]. The GC content (36.6–36.7%) also fell within the range reported in previous studies, ranging from 23.1% (*Gastrodia flexistyla*) to 37.8% (*Cypripedium macranthos*) [16,18].

In terms of gene content, *Trichoglottis* encoded a total of 120 genes, including 74 protein-coding genes, 38 transfer RNA (tRNA) genes, and 8 ribosomal RNA (rRNA) genes, with the pseudogenization or loss of all *ndh* genes (Figure 1, Table 1). The pseudogenization and loss of *ndh* genes were commonly observed in Orchidaceae [19,20,21]. It has been hypothesized that this might be associated with epiphytic habitats [22]. Epiphytic orchids usually undergo pseudogenization or loss of the *ndh* genes, such as *Dendrobium* [23], *Polystachya* [24], and Aeridinae [13,15]. For *Trichoglottis*, the results showed that four genes (*ndhA*/*F*/*H*/*I*) were completely lost and the other seven genes (*ndhB*/*C*/*D*/*E*/*G*/*J*/*K*) were pseudogenes; in *T*. *ionosma*, three genes (*ndhA*/*F*/*H*) were completely lost and the other eight genes (*ndhB*/*C*/*D*/*E*/*G*/*I*/*J*/*K*) were pseudogenes in the other six species. Species of *Trichoglottis* are usually epiphytic on tree trunks or lithophytic on rocks. Therefore, the results provide support for a close association between pseudogenization or loss of *ndh* genes and epiphytic habitats.

IR/SC boundary shift is a common phenomenon in the evolutionary process of angiosperms and serves as the main factor for the difference in plastome length and gene content [25,26]. In this study, the gene arrangement of the IR/SC boundary in seven *Trichoglottis* plastomes was extremely conserved; except for a slight difference at JSB, the *ycf1* in *T*. *latisepala* and *T*. *philippinensis* was entirely located within the IRb region, while in the remaining species, the *ycf1* crossed over JSB. These results indicated that no significant expansion or contraction occurred in the IR regions of *Trichoglottis*, which may be one of the primary factors contributing to the high conservation of plastome structure.

Simple sequence repeats (SSRs), widely distributed in plastomes, are effective molecular markers used for detecting genome rearrangement, species identification, and analyses of genetic differences among individuals [27,28]. A total of 98 (*T*. *philippinensis*) to 134 (*T*. *ionosma*) SSRs were detected, which is more than previous results [24,29,30,31]. This might be associated regions of increased nucleotide diversity [32]. Most SSRs were composed of A/T repeats rather than G/C repeats; this may be due to A/T having a more stable framework compared to G/C [33]. Additionally, hexanucleotide (AAGAAT/ATTCTT) was exclusively detected in the plastomes of *T*. *subviolacea*, making it a specific molecular marker for identifying *T*. *subviolacea*. The number of long repeats ranged from 33 (*T*. *subviolacea*) to 46 (*T*. *ionosma*), which is similar to previous studies [24,29,30]. The results contributed valuable molecular resources for the development of DNA markers in *Trichoglottis*.

Relative synonymous codon usage (RSCU) is an important measure to calculate the preference for synonymous codon usage. When the RSCU value is greater than 1, it indicates a preferred selection of that codon, whereas when it is less than 1, there is no preference [34]. The codon usage bias is highly conserved among plastomes of *Trichoglottis*. Leucine (Leu) had the highest amino acid frequency, while cysteine (Cys) had the lowest frequency. Knill et al. [35] proposed that the high leucine frequency may be attributed to the substantial demand for leucine in photosynthesis-related metabolism. When cysteine is accumulated above a specific threshold tolerated by the host, it can be considered toxic, resulting in the lowest abundance among amino acids [36]. Our results were consistent with previous results regarding codon preference in Orchidaceae [24,29,30,31].

### 3.2. The Barcoding Investigation and Phylogenetic Analysis

The phylogenetic relationship of *Trichoglottis* s.l. has remained unresolved for a long time. Based on nuclear ribosomal ITS and a number of plastid sequences, a conflicting relationship between *Staurochilus* and *Trichoglottis* s.s. was revealed, but weak support and unstable topology were found [6,7,8]. The genus *Trichoglottis* s.l. was strongly supported as a monophyletic group, and two species, *T*. *ionosma* and *T*. *philippinensis*, formerly treated as *Staurochilus*, were nested as successively basal lineages (Figure 4, Appendix A). The relationship was not consistent with previous studies based on traditional molecular markers [6,7,8], which could be due to the expanded sampling and markers with more parsimony-informative sites. However, our study also revealed extremely short branch lengths within *Trichoglottis* s.s. (Figure 4, Appendix A), indicating that our expanded taxonomic sample did not permit us to clarify the intrageneric relationships of *Trichoglottis* s.l. This phenomenon could be attributed to *Trichoglottis* s.l. having undergone rapid radiation, resulting in few opportunities for molecular changes due to the short time of speciation [37]. Therefore, future studies based on broader sampling and markers with different genetic backgrounds is needed.

Nucleotide diversity (Pi) was used to measure the degree of polymorphism. Similar to other Orchidaceae taxa, the non-coding regions of *Trichoglottis* plastomes had higher Pi values than the coding regions [24,29,31]. A total of five regions (*trnN^GUU^*-*rpl32* > *trnL^UAA^* > *trnS^GCU^*-*trnG^UCC^* > *rbcL*-*accD* > *trnT^GGU^*-*psbD*) were identified as mutation hotspots and could be used in the phylogenetic and identification of *Trichoglottis* and its related genera.

## 4. Materials and Methods

### 4.1. Taxon Sampling and Sequencing

Seven species of *Trichoglottis* were selected, among which five plastomes were newly sequenced in this study. Plant materials were collected from Fujian Agriculture and Forestry University (Fuzhou, Fujian Province, China). Based on previous molecular systematic studies [15], a total of 31 plastid genomes from 31 species were selected in this study, including six species from five genera (*Calanthe*, *Calypso*, *Cattleya*, *Masdevallia*, and *Tridactyle*) as the outgroups. The taxa with voucher information and GenBank accession numbers are provided in Appendix A.

Total DNA was extracted from fresh leaves with the Plant Mini Kit (Qiagen, CA. USA) based on the manufacturer’s protocol, and DNA degradation and contamination were evaluated on 1% agarose gels. Next-generation sequencing (NGS) was conducted using the Illumina Hiseq 4000 sequencing platform (Illumina, CA, United States), and 150-bp paired-end reads were produced. Scripts were used to filter the Illumina data in the cluster with the default parameter. When the low-quality (Q ≤ 5) base number in sequencing reads surpassed 50% of the reads base number and the N content in reads exceeded 10% of the reads base number, paired reads were eliminated from the analysis. More than 10 Gb of clean data were obtained for each species.

### 4.2. Plastome Assembly and Annotation

The paired-end sequencing reads were filtered and assembled into complete plastid genomes by using the GetOrganelle pipe-line [38] with default parameters. The obtained complete plastid genomes were annotated using PGA software (https://github.com/quxiaojian/PGA, accessed on 10 July 2023) [39], and the published plastome of *Trichoglottis philippinensis* (MN124404) was used as a reference. The annotated plastomes were manually examined using Geneious 11.1.5 [40]. Genes containing one or more internal stop codons, when compared to homologous genes, were classified as pseudogenes or partial copies. Genes with a ≥ 50% loss of the complete protein-coding genes or a similarity ≤ 50% were considered the lost genes. Finally, five high-quality complete plastid genomes were obtained. The annotation circle maps were drawn using OGDRAW v 1.3.1 [41].

### 4.3. Plastome Comparative Analysis

The Perl script MISA [42] was employed to detect simple sequence repeats (SSRs), with 10, 5, 4, 3, 3, and 3 nucleotide repeats set for mono-, di-, tri-, tetra-, penta-, and hexa-motif microsatellites (mononucleotide, dinucleotide, trinucleotide, tetranucleotide, pentanucleotide, and hexanucleotide) set as the minimum threshold, respectively. Four long repeat types in seven plastid genomes, F (forward), P (palindrome), R (reverse), and C (complement), were detected using REPuter [43] with default parameters. Results were visualized with the R package *ggplot2* [44]. The rearrangements between different *Trichoglottis* plastomes were identified by Mauve [45]. The IRscope online program [46] was used to compare the genes in the boundary regions of LSC/IRb/SSC/IRa.

### 4.4. Codon Usage Analysis

A total of 68 protein-coding genes from *Trichoglottis* plastomes were extracted using PhyloSuite v1.2.2 [47]. These protein-coding genes were concatenated using Phylosuite v1.2.2 and analyzed for the relative synonymous codon usage (RSCU) for each species of *Trichoglottis* using DAMBE [48]. Finally, a heatmap was generated using Tbtools v 1.120 [49].

### 4.5. Sequence Divergence and Barcoding Investigation

The online program mVISTA was used to analyze the diversity of plastome sequences of *Trichoglottis* using the Shuffle-LAGAN [50] alignment program. The *Thrixspermum centipeda* (MW057769) plastome was used as a reference. The nucleotide variability (Pi) of complete plastomes and 68 protein-coding genes of seven *Trichoglottis* species was calculated using DnaSP 6 [51] with a window length of 100 sites and a step size of 25 sites.

### 4.6. Phylogenetic Reconstruction

In this study, phylogenetic trees were constructed using a concatenated matrix of 68 protein-coding genes and a matrix of complete plastomes. The 68 protein-coding genes (*ndh* genes were widespread, pseudogenized, or lost in Aeridinae species) were extracted using PhyloSuite v1.2.2 [47] and aligned using MAFFT [52]. The aligned 68 protein-coding genes were concatenated using PhyloSuite v1.2.2 [47]. The complete plastid genomes were aligned by MAFFT, and TrimAL v1.4 was employed [53] to trim the poorly aligned positions with the default parameter.

The phylogenetic trees were inferred by maximum likelihood (ML), maximum parsimony (MP), and Bayesian inference (BI) on the website CIPRES Science Gateway web server (RAxML-HPC2 on XSEDE 8.2.12, PAUP on XSEDE 4.a 168, and MrBayes on XSEDE 3.2.7a) [54]. For ML analysis, the GTRGAMMA model was specified for all datasets [55] and bootstrap values were calculated from 1000 bootstrap replicates using heuristic searches [56]. For BI analysis, we used MrBayes v. 3.2.7a under the GTR + I + Γ substitution model. The Markov chain Monte Carlo (MCMC) algorithm was run for 10,000,000 generations, with one tree sampled every 100 generations. The first 25% of trees were discarded as burn-in to construct majority-rule consensus trees and estimate posterior probabilities (PPs).

## 5. Conclusions

In this study, five plastomes of *Trichoglottis* (*T*. *cirrhifera*, *T*. *ionosma*, *T*. *lanceolaria*, *T*. *orchidea*, and *T*. *subviolacea*) were newly assembled. Our results showed that the structure and gene content of *Trichoglottis* plastomes are highly conserved. All *ndh* genes were lost or pseudogenized. The *Trichoglottis* s.l. (including *Staurochilus*) was strongly supported as a monophyletic group, and two species formerly treated as *Staurochilus* were revealed as successively basal lineages. Five non-coding regions (*trnN^GUU^*-*rpl32* > *trnL^UAA^* > *trnS^GCU^*-*trnG^UCC^* > *rbcL*-*accD* > *trnT^GGU^*-*psbD*) were identified to further elucidate the phylogeny of *Trichoglottis*. These findings provide valuable insights into plastome evolutionary patterns and phylogenetic relationships for *Trichoglottis* and its related genera and even extend to the orchid family.

## Figures and Tables

**Figure 1 ijms-24-14544-f001:**
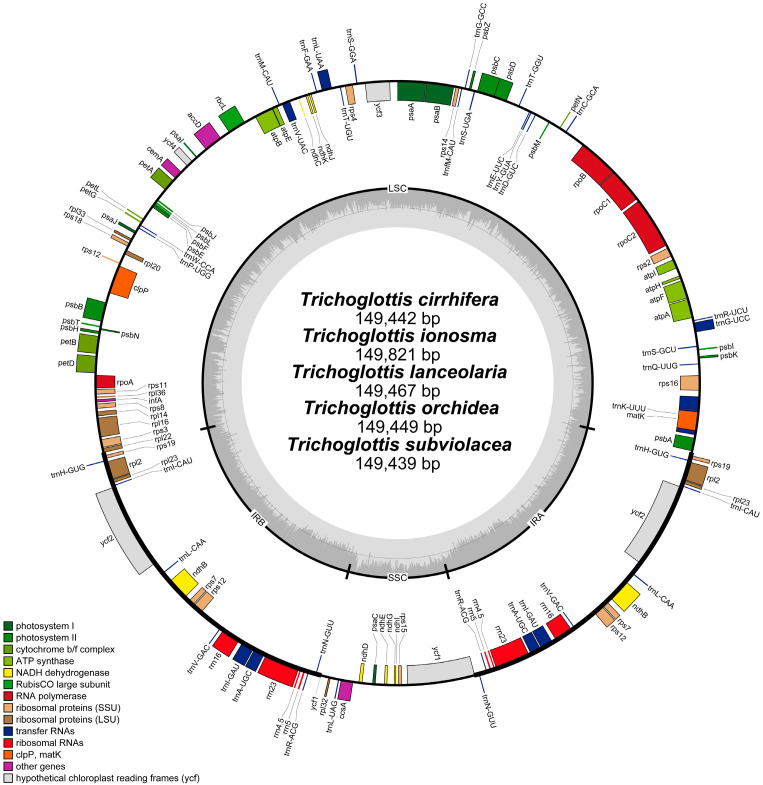
The plastome annotation map of *Trichoglottis cirrhifera*, *Trichoglottis ionosma*, *Trichoglottis lanceolaria*, *Trichoglottis orchidea*, and *Trichoglottis subviolacea*. The darker gray in the inner circle corresponds to GC content. The IRA and IRB (two inverted repeating regions); LSC (large single-copy region); and SSC (small single-copy region) are indicated outside of GC content.

**Figure 2 ijms-24-14544-f002:**
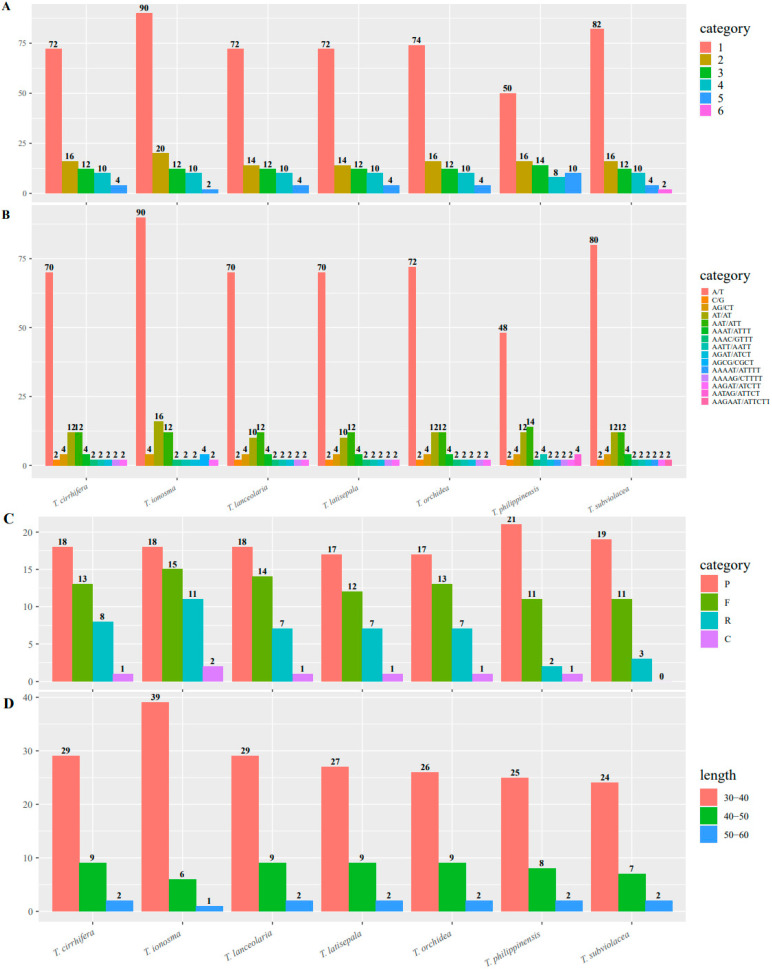
Summary of simple sequence repeats (SSR) across the *Trichoglottis* plastomes. (**A**) Frequency of identified SSR motifs (mono-, di-, tri-, tetra-, penta-, and hexa-). (**B**) Frequency of classified repeat types (considering sequence complementary). (**C**) Variation in repeat abundance and type. (**D**) Number of long repeats by sequence length.

**Figure 3 ijms-24-14544-f003:**
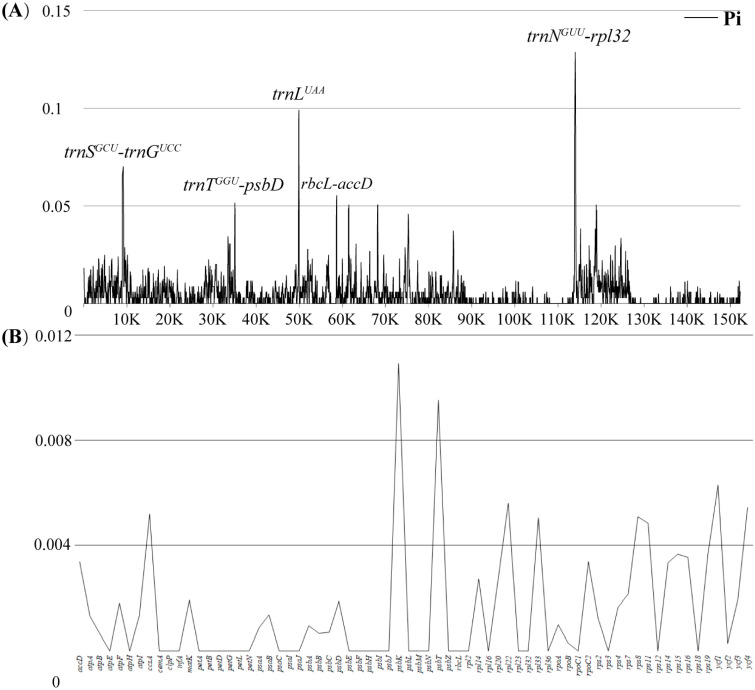
Sliding window test of nucleotide diversity for *Trichoglottis* plastomes. (**A**) The nucleotide diversity of the complete plastome; five mutation hotspot regions were annotated. (**B**) The nucleotide diversity of 68 protein-coding sequences. The window size was set to 100 bp, and the sliding window size was 25 bp. *X*-axis: position of the midpoint of a window; *Y*-axis: nucleotide diversity of each window or gene.

**Figure 4 ijms-24-14544-f004:**
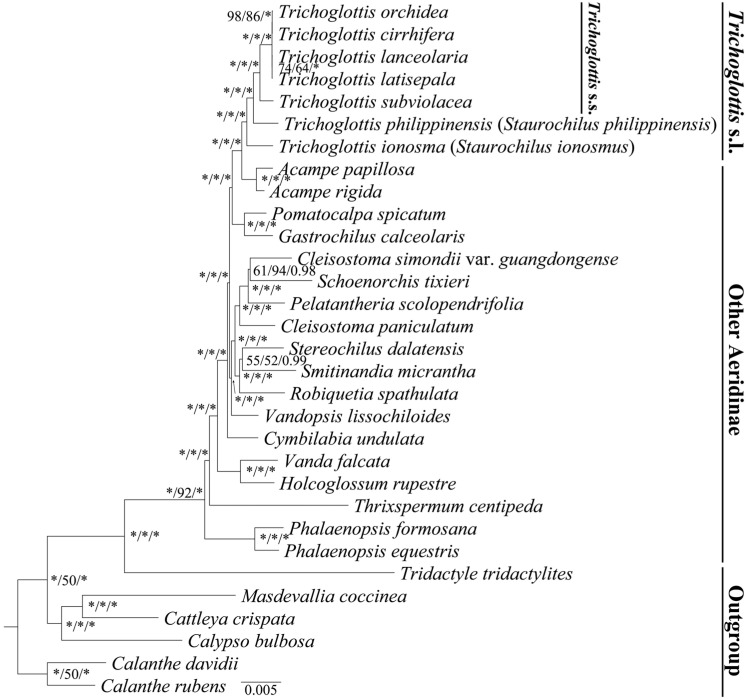
Phylogenetic tree obtained by maximum-likelihood analysis based on complete plastome. The numbers near the nodes are bootstrap percentages and Bayesian posterior probabilities (BP_ML_, BP_MP_, PP). A dash (-) indicates that a node is inconsistent between the topology of the MP/ML trees and the Bayesian tree; * node is 100 bootstrap percentage or 1.00 posterior probability.

**Table 1 ijms-24-14544-t001:** Characteristics of the complete plastomes of *Trichoglottis*. An asterisk (*) indicates those newly sequenced in this study.

Species Name	Size (bp)	GC Content (%)	LSC Size in bp (%)	IR Size in bp (%)	SSC Size in bp (%)	Total Number of Gene	Protein-Encoding Gene	tRNA	rRNA	Number of *ndh* Fragment
*T. cirrhifera* *	149,442	36.7	85,721 (57.36)	25,775 (17.25)	12,171 (8.14)	120	74	38	8	8
*T*. *ionosma* *	149,841	36.6	86,210 (57.53)	25,812 (17.23)	12,007 (8.01)	120	74	38	8	7
*T*. *lanceolaria* *	149,467	36.7	85,728 (57.36)	25,784 (17.25)	12,171 (8.14)	120	74	38	8	8
*T*. *latisepala*	149,402	36.7	85,681 (57.35)	25,784 (17.26)	12,153 (8.13)	120	74	38	8	8
*T*. *orchidea* *	149,449	36.7	85,715 (57.35)	25,782 (17.25)	12,170 (8.14)	120	74	38	8	8
*T*. *philippinensis*	149,663	36.7	86,069 (57.51)	25,784 (17.23)	12,026 (8.04)	120	74	38	8	8
*T*. *subviolacea* *	149,439	36.7	85,853 (57.45)	25,791 (17.26)	12,004 (8.03)	120	74	38	8	8

## Data Availability

All the data is provided within this manuscript.

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
