# Peer review of "Characteristics and Comparative Analysis of Seven Complete Plastomes of Trichoglottis s.l. (Aeridinae, Orchidaceae)"

_ijms, 2023, doi:10.3390/ijms241914544_

Round 1

Reviewer 1 Report

The manuscript by Cheng-Yuan Zhou and colleagues (Comparative and phylogenetic analysis for Trichoglottis (Aeridinae, Orchidaceae) based on seven complete plastid genomes) is a well done work using standard methods and bioinformatic tools. As such, I think it should be accepted for publication, after the following corrections have been done. 

line 145 -"..the mVISTA (software) was employed" instead of ".. the mVISTA were employed".

line 189 - It's true you have expanded the plastome sampling of Trichoglottis, but please expand a bit your conclusions in this regard.  

I have a doubt with your election of outgroup in the plastome divergence analysis. Acampe rigida would be too close to Trichoglottis as to show changes or rearrangements in the cp genome. As often is the case in phylogenetic studies the better resolution is achieved by means of the use of at least two outgroups (as you have done in your reconstruction of page 7), it would be interesting to see if there is divergence within the Aeridinae. I would suggest re doing the mVISTA using as outgroup some of the genera you used in your phylogenetic reconstruction e.g. "other Aeridinae" or "outgroup". Please be sure to label the bars corresponding to each species correctly. In the figure S4 the only readable names are Trichoglottis philippinensis and T. subviolacea. Names of markers or genes are neither clear enough. 

Author Response

General Comments: The manuscript by Cheng-Yuan Zhou and colleagues (Comparative and phylogenetic analysis for Trichoglottis (Aeridinae, Orchidaceae) based on seven complete plastid genomes) is a well done work using standard methods and bioinformatic tools. As such, I think it should be accepted for publication, after the following corrections have been done.

Response: Thanks for your comments. We have revised the problems. Please refer following response.

Comment 1. line 145 -"..the mVISTA (software) was employed" instead of ".. the mVISTA were employed".

Response: Thank you very much. The ".. the mVISTA were employed" have been revised to "..the mVISTA (software) was employed". Please see page 5 line 147.

Comment 2. line 189 - It's true you have expanded the plastome sampling of Trichoglottis, but please expand a bit your conclusions in this regard.

Response: Thank you very much. We have added the content according to your kind advice. Please see page 8 lines 253-254.

Comment 3. I have a doubt with your election of outgroup in the plastome divergence analysis. Acampe rigida would be too close to Trichoglottis as to show changes or rearrangements in the cp genome. As often is the case in phylogenetic studies the better resolution is achieved by means of the use of at least two outgroups (as you have done in your reconstruction of page 7), it would be interesting to see if there is divergence within the Aeridinae. I would suggest re doing the mVISTA using as outgroup some of the genera you used in your phylogenetic reconstruction e.g. "other Aeridinae" or "outgroup". Please be sure to label the bars corresponding to each species correctly.

Response: Thank you very much. We have replaced the outgroup of the mVISTA according to your kind advice. Please see the Supplementary Figure S4, page 5 line 149 and page 9 line 313.

Comment 4. In the figure S4 the only readable names are Trichoglottis philippinensis and T. subviolacea. Names of markers or genes are neither clear enough.

Response: Thank you very much. We have adjected the figures to ensure that all marker and gene names are clear. Please see the Supplementary Figure S4.

Reviewer 2 Report

Dear Authors,

1. Although the topic of the study is interesting, already multiple researches have published the topic with almost the same results. Unfortunately, your study does not bring any novelty. You should give evidence that your study is original and is not only a repetition of what has been done and achieved.

2. I recommend the authors discuss and compare their findings with what has been achived earlier by other researchers.

3. All materials and methods should be described in detail and not in brief.

4. All scientific names should be italicized including the what is written in the paper title.

5. I recommend the authors double-check the full text for grammatical and typing errors.

I recommend the authors double-check the full text for grammatical and typing errors.

Author Response

Comment 1. Although the topic of the study is interesting, already multiple researches have published the topic with almost the same results. Unfortunately, your study does not bring any novelty. You should give evidence that your study is original and is not only a repetition of what has been done and achieved.

Response: Thank you very much. We have made significant revisions to our manuscript according to your kind advice. The details please see the manuscript, especially in Abstract, Introduction and Conclusion.

Comment 2. I recommend the authors discuss and compare their findings with what has been achived earlier by other researchers.

Response: Thank you very much. We have made significant revisions to our manuscript according to your kind advice. Please see Introduction and Discussion.

Comment 3. All materials and methods should be described in detail and not in brief.

Response: Thank you very much. We have added more detailed content about materials and methods according to your kind advice. Please see page 9-10.

Comment 4. All scientific names should be italicized including the what is written in the paper title.

Response: Thank you very much. We have ensured that all scientific names are properly italicized in the revised manuscript.

Comment 5. I recommend the authors double-check the full text for grammatical and typing errors.

Response: Thank you very much. We have modified and improved the English by the native speaker.

Reviewer 3 Report

Here are several major concerns:

- The introduction lacks depth and does not sufficiently set the stage for why this research is needed. There are large gaps in describing the background context and motivation for the study. The objectives require clearer framing.

- The methodology requires extensive additional details and justification. The experimental design and analysis process is not explained adequately. Many steps are unclear or missing.

- The results are presented in a very disjointed manner that is difficult to follow logically. There is insufficient interpretation and discussion connecting the different findings. The writing quality needs significant improvement.

- The presented data does not strongly support the conclusions made. The claims reach beyond what the data demonstrates. Additional experiments and analyses appear needed.

- The phylogenetic relationships revealed contradict some previous studies, but no plausible explanations are provided for why this occurred. More in-depth comparative analysis is needed.

- The discussion and conclusions are very superficial and do not place the findings in the context of the broader literature. The implications are not sufficiently addressed.

- Throughout the paper, there are issues with grammar, awkward wording, formatting inconsistencies, and undefined terms/acronyms. Extensive editing is required.

- The contributions to knowledge are incremental at best and not clearly articulated. The innovations of this study compared to prior work require elaboration.

In summary, I think the manuscript has pervasive flaws in the framing, methodology, data analysis, results presentation, and writing quality. I would not recommend it for publication without undergoing major revisions to address these issues. Please feel free to contact me if you would like any clarification or expansion on my comments.

Spell Check Required.

Author Response

Comment 1.- The introduction lacks depth and does not sufficiently set the stage for why this research is needed. There are large gaps in describing the background context and motivation for the study. The objectives require clearer framing.

Response: Thank you very much. We have made significant revisions to our manuscript, including Abstract and Introduction, according to your kind advice.

Comment 2.- The methodology requires extensive additional details and justification. The experimental design and analysis process is not explained adequately. Many steps are unclear or missing.

Response: Thanks very much. We have made significant revisions to provide a more detailed and comprehensive explanation of our experimental design and analysis process in Materials and Methods. Please see page 9-10.

Comment 3.- The results are presented in a very disjointed manner that is difficult to follow logically. There is insufficient interpretation and discussion connecting the different findings. The writing quality needs significant improvement.

Response: Thank you very much. We have reworded the results. Please see the manuscript.

Comment 4.- The presented data does not strongly support the conclusions made. The claims reach beyond what the data demonstrates. Additional experiments and analyses appear needed.

Response: Thank you very much. We have reworded the Introduction and Conclusion. Please see the manuscript.

Comment 5.- The phylogenetic relationships revealed contradict some previous studies, but no plausible explanations are provided for why this occurred. More in-depth comparative analysis is needed.

Response: Thank you very much. We have provided more detailed comment in discussion. Please page 9 line 290-302.

Comment 6.- The discussion and conclusions are very superficial and do not place the findings in the context of the broader literature. The implications are not sufficiently addressed.

Response: Thank you very much. We have reworded our introduction, discussion and conclusions. Please see the manuscript pages 7, 8, 9 and 10.

Comment 7.- Throughout the paper, there are issues with grammar, awkward wording, formatting inconsistencies, and undefined terms/acronyms. Extensive editing is required.

Response: Thank you very much. We have improved the English through the manuscript.

Comment 8.- The contributions to knowledge are incremental at best and not clearly articulated. The innovations of this study compared to prior work require elaboration.

Response: Thank you very much. We have improved the knowledge and the innovations have been revised in introduction and discussion.

Comment 9. In summary, I think the manuscript has pervasive flaws in the framing, methodology, data analysis, results presentation, and writing quality. I would not recommend it for publication without undergoing major revisions to address these issues. Please feel free to contact me if you would like any clarification or expansion on my comments.

Response: Thank you for your comments. We have made significant revisions to our manuscript, especially in Abstract, Introduction and Conclusion. We hope that these improvements address your concerns.

Reviewer 4 Report

The authors investigated the phylogenetic relationships of Trichoglottis based on complete plastome sequences from seven species of this genus. The authors have also comprehensively characterized the plastome structures of these 7 taxa.

I believe that the overall presentation is fine with acceptable language (English). But I have one major concern about this study. The title of article reflects and emphasizes the phylogenetic analysis of the genus Trichoglottis. However, the authors never indicated how many taxa this genus contains. This is somehow misleading because it is not a sufficient sampling strategy to include only 7 species out of several dozens of taxa in this genus. The authors indicated one of the aims was to resolve the phylogenetic relationships of the genus Trichoglottis. Again, this goal cannot be adequately achieved with such a small sample size. As a matter of fact, the four groups of the 7 taxa of Trichoglottis circumscribed in the current study are rather preliminary without a comprehensive sampling and subsequently, no meaningful discussion of the phylogenetic relationships in the genus Trichoglottis was provided in the Discussion. To me, it makes more sense to say that this is a phylogenetic study of the subtribe Aeridinae, rather than the genus Trichoglottis, based on a total of 26 taxa.

Here is the solution to this problem that I would like to propose:

It is still good to keep the characterizations of the plastome of these newly sequenced taxa. However, the authors should revise the wording in title, aim of this study, more background othe phylogenetic relationships in the subtribe Aeridinae, and more importantly, the in-depth discussion to focus on the phylogenetic relationships of the subtribe Aeridinae, rather than the genus Trichoglottis. I believe that these suggestions are legitimate and should significantly help improve the manuscript, if a revision is requested by the editor.

Minor revisions:

Line 69: the first part of this sentence “This study aim to examine and document with …” reads strange, please correct it.

Line 85: As a representative, Figure 1 should be based on one specific species, which species is it?

The English is acceptable.

Author Response

General Comments: The authors investigated the phylogenetic relationships of Trichoglottis based on complete plastome sequences from seven species of this genus. The authors have also comprehensively characterized the plastome structures of these 7 taxa. I believe that the overall presentation is fine with acceptable language (English). But I have one major concern about this study.

Response: Thanks for your comments. We have made significant revisions to our manuscript according to your kind advice. Please see the manuscript and refer following responses.

Comment 1. The title of article reflects and emphasizes the phylogenetic analysis of the genus Trichoglottis. However, the authors never indicated how many taxa this genus contains. This is somehow misleading because it is not a sufficient sampling strategy to include only 7 species out of several dozens of taxa in this genus. The authors indicated one of the aims was to resolve the phylogenetic relationships of the genus Trichoglottis. Again, this goal cannot be adequately achieved with such a small sample size. As a matter of fact, the four groups of the 7 taxa of Trichoglottis circumscribed in the current study are rather preliminary without a comprehensive sampling and subsequently, no meaningful discussion of the phylogenetic relationships in the genus Trichoglottis was provided in the Discussion. To me, it makes more sense to say that this is a phylogenetic study of the subtribe Aeridinae, rather than the genus Trichoglottis, based on a total of 26 taxa.

Response: Thank you very much. We have made significant revisions to our manuscript according to your kind advice. Please see Title, Abstract and Instruction.

Comment 2. Here is the solution to this problem that I would like to propose:

It is still good to keep the characterizations of the plastome of these newly sequenced taxa. However, the authors should revise the wording in title, aim of this study, more background othe phylogenetic relationships in the subtribe Aeridinae, and more importantly, the in-depth discussion to focus on the phylogenetic relationships of the subtribe Aeridinae, rather than the genus Trichoglottis. I believe that these suggestions are legitimate and should significantly help improve the manuscript, if a revision is requested by the editor.

Response: Thank you very much. We have revised the title and made the necessary adjustments to the introduction and discussion sections to ensure a more comprehensive exploration about Trichoglottis. Please see the manuscript.

Comment 3. Line 69: the first part of this sentence “This study aim to examine and document with …” reads strange, please correct it.

Response: Thanks for your useful suggestions. We have reworded this part. Please see page 2.

Comment 4. Line 85: As a representative, Figure 1 should be based on one specific species, which species is it?

Response: Thank you very much. Given the highly similar plastome structure among Trichoglottis, we used the annotation map of T. cirrhifera as a representative and all the other four newly sequenced species were presented.

Round 2

Reviewer 2 Report

The manuscript has been sufficiently improved.

Author Response

Comment 1. The manuscript has been sufficiently improved.

Response: Thank you very much. 

Reviewer 4 Report

I appreciate very much the authors’ effort devoted to improving their manuscript. However, I am now concerned about the authors’ response to my comments, quoted from my previous review,

“Here is the solution to this problem that I would like to propose:

It is still good to keep the characterizations of the plastome of these newly sequenced taxa. However, the authors should revise the wording in title, aim of this study, more background othe phylogenetic relationships in the subtribe Aeridinae, and more importantly, the in-depth discussion to focus on the phylogenetic relationships of the subtribe Aeridinae, rather than the genus Trichoglottis. I believe that these suggestions are legitimate and should significantly help improve the manuscript, if a revision is requested by the editor.”

The authors responded my above comments, “We have revised the title and made the necessary adjustments to the introduction and discussion sections to ensure a more comprehensive exploration about Trichoglottis. Please see the manuscript.”

However, I do not see any significant change at all in Discussion, i.e., Section 3.2.

I believed that my suggestions were fair and will significantly improve the manuscript. If the authors do not want to expand on the ibn-depth discussion of the phylogenetic investigations, then, I would propose another easier solution to this problem: I would highly recommend that the authors change the title to emphasize the significance of the characterization of these genomes because this study, with its contents considered, is not really focusing on the phylogenetic study at all, which is rather just a by-product of these genomes. Again, I believe that this would work as well so that the contents of this manuscript fit its title.

Also, please indicate the species name, but not just the genus name, in the legend of Figure 1.

The English is acceptable.

Author Response

Comment 1. I appreciate very much the authors’ effort devoted to improving their manuscript. However, I am now concerned about the authors’ response to my comments, quoted from my previous review,

“Here is the solution to this problem that I would like to propose:

It is still good to keep the characterizations of the plastome of these newly sequenced taxa. However, the authors should revise the wording in title, aim of this study, more background othe phylogenetic relationships in the subtribe Aeridinae, and more importantly, the in-depth discussion to focus on the phylogenetic relationships of the subtribe Aeridinae, rather than the genus Trichoglottis. I believe that these suggestions are legitimate and should significantly help improve the manuscript, if a revision is requested by the editor.”

The authors responded my above comments, “We have revised the title and made the necessary adjustments to the introduction and discussion sections to ensure a more comprehensive exploration about Trichoglottis. Please see the manuscript.”

However, I do not see any significant change at all in Discussion, i.e., Section 3.2.

I believed that my suggestions were fair and will significantly improve the manuscript. If the authors do not want to expand on the ibn-depth discussion of the phylogenetic investigations, then, I would propose another easier solution to this problem: I would highly recommend that the authors change the title to emphasize the significance of the characterization of these genomes because this study, with its contents considered, is not really focusing on the phylogenetic study at all, which is rather just a by-product of these genomes. Again, I believe that this would work as well so that the contents of this manuscript fit its title.

Response: Thank you very much. We have reworded the Title according to your kind advice. Please see the manuscript.

Comment 2. Also, please indicate the species name, but not just the genus name, in the legend of Figure 1.

Response: Thank you very much. We have reworded. Please see line 89-90.